# The Oral Microbiome Profile of Water Polo Players Aged 16–20

**DOI:** 10.3390/sports11110216

**Published:** 2023-11-07

**Authors:** Irina Kalabiska, Dorina Annar, Zsuzsa Keki, Zoltan Borbas, Harjit Pal Bhattoa, Annamaria Zsakai

**Affiliations:** 1Research Center for Sport Physiology, Hungarian University of Sports Science, Alkotas u. 44, 1123 Budapest, Hungary; kalabiskai@gmail.com (I.K.); annar.dorina@gmail.com (D.A.); borbasmano@gmail.com (Z.B.); 2Doctoral School of Biology, Eotvos Lorand University, Pazmany P. s. 1/c, 1117 Budapest, Hungary; 3Biomi Ltd., Szent-Gyorgyi Albert u. 4, 2100 Godollo, Hungary; keki.zsuzsa@biomi.hu; 4Department of Laboratory Medicine, Faculty of Medicine, University of Debrecen, Nagyerdei Blvd. 98, 4032 Debrecen, Hungary; harjit@med.unideb.hu; 5Department of Biological Anthropology, Eotvos Lorand University, Pazmany P. s. 1/c, 1117 Budapest, Hungary; 6Health Promotion and Education Research Team, Hungarian Academy of Sciences, 1117 Budapest, Hungary

**Keywords:** young water polo players, oral microbiome, DADA2 library, sexual dimorphism

## Abstract

Objectives: Chlorine has a strong antibacterial property and is the disinfectant most frequently used in swimming pools. Therefore, the microbiota community in the oral cavity of those who practice water sports is assumed to be special due to their regular immersion in water. Adverse changes in the composition of oral cavity microbiota may have serious health consequences. We aimed to compare the oral microbiome between water polo players and non-athletes. We hypothesized that the oral cavity microbiota community differed between water polo players and non-athletes. Materials and Methods: Altogether, 124 water polo players (62 males and 62 females, aged between 9 and 20 years) and 16 non-athlete youths (control group, eight males and eight females, aged between 16 and 20 years, mean age + SD = 17.1 + 1.4 years) who participated in body structure examinations voluntarily agreed to participate in the study. In a randomly selected subsample of water polo players (*n*: 29, aged between 16 and 20 years, mean age + SD = 17.3 + 1.0 years), saliva samples were also collected. Saliva samples were collected from all non-athlete youths (*n*: 16, aged between 16 and 20 years). The oral microbiome was determined from a saliva sample, and DNA was isolated using the QIAmp DNA Blood Mini Kit. The 16S rRNA gene amplicon sequencing method was used to analyze the microbiome community. PCR primers were trimmed from the sequence reads with Cutadapt. R library DADA2 was used to process reads in the abundance analysis. Results: In general, *Streptococcus*, *Veilonella*, and *Prevotella* genera constituted more than 50% of the oral microbiome community in the two participant groups combined (*n* = 45). The oral microbial profile had significant sexual dimorphism and differed between water polo players and the non-athletes. Compared to females, males had a higher (*p* < 0.05) relative abundance of the *Atopobium* (medium effect size) and *Pravotella*_7 (very large effect size) genera and a lower (*p* < 0.05) relative abundance of the *Fusobacterium* (large effect size), *Gemella* (large effect size), and *Streptococcus* (large effect size) genera. Compared to non-athletes, water polo players had higher (*p* < 0.05, medium effect size) relative abundance of the genus *Veillonella* and lower (*p* < 0.05, large effect size) relative abundance of the genus Gemella. Conclusions: The results suggest that regular water training can unfavorably alter the composition of the oral microbial community.

## 1. Introduction

Previous studies of the human oral microbiome have identified viruses, fungi, protozoa, archaea, and bacteria in the community of microorganisms that colonize the surfaces of teeth and the soft tissues of the oral mucosa [1]. The oral microbiome is an ecological community of symbiotic, commensal, and pathogenic microorganisms. The oral microbiome comprises 500–700 bacterial species, while 3000–7000 discernible operational taxonomic unit-level (OTU, used to differentiate bacterial organisms below the genus level) phylotypes were present in the bacterial community of the oral cavity. The most frequently identified salivary bacterial phyla are Actinobacteria, Bacteroidetes, Firmicutes, Fusobacteria, and Proteobacteria [2,3]. The oral microbiome has important roles in maintaining oral homeostasis, protecting the oral cavity, preventing the evolution of oral diseases arising from food digestion, generating energy, promoting the maturation of the host’s mucosa and immune system, controlling fat storage and metabolic regulation, detoxicating environmental chemicals, and preventing the invasion of pathogens [3]. Any changes in the microbiota community of the oral cavity can affect health. Such changes could include, but are not limited to, an altered metabolism of the microbial community or the host or changes in the factors affecting the biofilm in the oral cavity. In dysbiosis, when the equilibrium of the oral ecosystem is disrupted, the disease-promoting bacteria manifest and lead to deteriorating health.

In addition to the significant influence of host genetics on the composition of the salivary microbiome, it has been distinctly observed that lifestyle factors play an even more substantial role in shaping the diversity of the salivary microbiome [4]. Factors such as age, gender, body composition, ethnic background, geographical location, and sociodemographic variables as well as habits such as smoking and alcohol consumption have all been demonstrated to exert a discernible impact on the composition of the oral microbiome [5,6,7,8,9,10,11,12].

Regular exercise can favorably modify human organ systems’ functions and reduce the risk of obesity and insulin resistance and improve skeleton-muscular robusticity, stress tolerance, and immune function. However, while mild-to-moderate intensity exercises can protect against chronic disease, acute strenuous exercise can lead to injuries and illness in athletes. For example, overuse injuries can develop in swimmers’ upper extremities, or gastrointestinal tract symptoms and gut ischemia can cause nausea, vomiting, abdominal pain, and diarrhea in runners [13,14]. The research carried out by Tripodi et al. has also confirmed that among athletes participating in various sports, engaging in sports activities can be regarded as a potential risk factor for the development of oral diseases [15].

Water polo players are at risk for developing traumatic injuries due to intense physical contact as well as overuse injuries due to repetitive movements against resistance. In addition, water polo players often experience otitis, skin allergic issues, eye irritation, asthma, and exercise-induced bronchospasm due to the aquatic environment and prolonged exposure to chlorine [16]. The effects of the aquatic environment on the microorganism communities in water polo players’ gastrointestinal tract (oral and feces microbiomes) have not yet been examined. A lack of such data is surprising because gastrointestinal tract infections occur at 1.5–3.0 times higher frequency in water polo players compared with non-athletes, second after upper respiratory tract infections [14,17,18,19]. Indeed, several studies reported that water polo players are exposed to high levels of gastrointestinal tract pathogens while in the water [20,21]. *Cryptosporidium*, *Giardia intestinalis*, and *Adenovirus* strains 4 and 7 are the most common cause of swimming pool-related gastrointestinal illness, as these pathogens are partially chlorine-resistant [22].

Chlorine is the most common disinfectant used in swimming pools and can irritate or disturb the microbial defense system of the normal body. Chlorine modifies the composition of the microbiota community in the large intestine and the growth of pathogenic microbes and can eventually affect the performance of athletes. The microbiota community in water polo players’ oral cavity is assumed to be unique due to the prolonged time these athletes spend in water. We assume that, compared to the community composition corresponding to their age group, we may observe differences in the dominance of the microbial strains. The composition of the oral microbiota community changes continuously and dynamically from birth to old age, but its changes are most intensively demonstrated during the period of physical development.

Hydration habits can potentially affect the composition of the oral microbiome. The oral microbiome is a complex community of microorganisms, including bacteria, viruses, fungi, and other microorganisms, that inhabit the oral cavity. These microorganisms play a crucial role in maintaining oral health and can influence overall health as well. Hydration status directly affects saliva production. Saliva contains antimicrobial proteins and enzymes that help control the growth of harmful bacteria in the mouth. When you are well hydrated, your salivary flow is typically normal, which can contribute to a healthier oral microbiome [23]. Saliva helps maintain the pH balance in the mouth. Adequate hydration can support the buffering capacity of saliva, preventing the oral environment from becoming too acidic or too alkaline, which can impact the types of microorganisms that thrive in the mouth [24]. Dehydration can lead to dry mouth (xerostomia), which can be a breeding ground for harmful bacteria. When there is insufficient saliva to wash away food particles and neutralize acids, it can lead to an imbalance in the oral microbiome, allowing harmful bacteria to flourish [25,26]. Water polo players as well as swimmers might be characterized by peculiar water intake during training due to the characteristics of their specific sport. There was a wide individual variation in fluid intake and sweat loss in both water polo players and swimmers [27]. Dehydration experienced by athletes in that study was less than typically reported for “land-based” athletes.

Unlike other team sports, few researchers have examined the water intake habits of water polo athletes which affect the athletes’ hydration. Overall, while hydration habits alone may not determine the composition of the oral microbiome, they can play a significant role in maintaining a healthy oral environment. Staying well hydrated, practicing good oral hygiene, and making healthy dietary choices can all contribute to a balanced and beneficial oral microbiome.

The purpose of the study was to compare the oral microbiome between water polo players and non-athletes. We hypothesized that the oral cavity’s microbiota community differed between water polo players and non-athletes due to training in chlorinated water.

## 2. Subjects and Methods

### 2.1. Subjects

Altogether, 124 water polo players (62 males and 62 females, aged between 9 and 20 years) and 16 non-athlete youths (control group, 8 males and 8 females, aged between 16 and 20 years, mean age + SD = 17.1 + 1.4 years) who participated in body structure examinations voluntarily agreed to participate in the study. In a randomly selected subsample of water polo players (*n*: 29, aged between 16 and 20 years, mean age + SD = 17.3 + 1.0 years), saliva samples were also collected. Saliva samples were collected from all non-athlete youths (*n*: 16, aged between 16 and 20 years).

### 2.2. Methods

#### 2.2.1. Study Design

This was a cross-sectional study conducted in November 2022. Participants were asked to avoid eating and drinking for at least 30 min prior to examination and follow the habitual training regime during the week of the examination. The assessments took place between 9:00 and 12:00 in the morning.

Inclusion criteria of the participants were: age; gender; training regime (in the case of the control group, an inactivity time); health status; medication use; time since last food or drink; smoking abuse; specific medical conditions; pregnancy status.

Exclusion criteria of the participants were: communicable diseases (HIV, hepatitis, or COVID-19); recent dental work; medications that alter saliva (anticholinergic medications); serious medical conditions; drug abuse; allergies or sensitivities; inability to provide consent; participation in other studies; ethical considerations.

Participants provided written informed consent before the start of the measurements. The local ethics committee approved the study (TE-KEB/No42/2019), which was conducted according to the most recent version of the Declaration of Helsinki.

#### 2.2.2. Microbial Analyses

Saliva samples were collected with commercially available kits. Subjects were asked to avoid eating, drinking, chewing gum, or brushing their teeth for 30 min before sampling. Saliva samples of 500 µL were collected during the anthropometric examinations in water polo players and in schools that control participants attended. The saliva samples were stored at −20 °C for no longer than 4 weeks before microbiome analyses. These samples were used for DNA extraction, which was isolated by the QIAmp DNA Blood Mini Kit (Qiagen, Beverly, MA, USA). We used the 16S rRNA gene amplicon sequencing (16S analysis) method (by V3–V4 primer set) to analyze the microbiome community [28]. PCR primers were trimmed from the sequence reads with Cutadapt (Version 3.5).

R library DADA2 (Version 1.24.0) was used to process reads in the abundance analysis process. DADA2 trimming process was used to trim read segments after 0 nucleotide quality score was detected. Reads shorter than 196 nucleotides were filtered out. The maximum expected erroneous nucleotides were limited to 2 in the forward reads and 2 in the reverse reads. Reads with higher number of expected erroneous nucleotides were filtered out. Forward and reverse reads were merged if the overlapped region was at least 12 nucleotides long. In the overlapping region, max 0 mismatch was enabled. The minimum length of merged amplicon sequence variant was set to 390.

In the taxonomy assignment, both orientations of the merged reads were used to find best hit to the Silva 16S v138.1 reference sequence database on minimum bootstrap value level of 75. During the DADA2 workflow, 11,202 unique amplicon sequence variants were found in 45 samples. Out of the 11,202 amplicon sequence variants, the process identified 8361 chimera sequences and 116 amplicon sequence variants that were shorter than the minimum amplicon length, set at 390 nucleotides.

#### 2.2.3. Statistical Analyses

Principal coordinate (PCo) analysis of the bacterial composition data was used to visualize the microbiome profile. The ordination plot is a graphical representation of the relationship between sample compositions (e.g., taxa diversities) and helps to recognize patterns in the sample relationship or help to recognize environmental variables that may affect sample diversity. Samples with similar features may be located in a distinct area of the plot if the feature has some significant effect on the taxonomy diversity. In the principal coordinate analysis, the distance matrix was transformed into a new set of orthogonal axes—Axis.1 and Axis.2—that can explain the maximum amount of variation present in the dataset by descending order. The axis labels represent the % of variability explained by each PCo axis in the figures.

An ordination statistical test was used to compare distances of samples within the same group to distances of samples from different groups. If the distance between samples from the different groups was much larger than samples from the same group, we conclude that the groups were not equal. Therefore, the compositional similarities between the different groups were studied with permutational multivariate analysis of variance (PERMANOVA) and, for differential abundance testing, univariate analysis of relative abundance of genera was performed with ANOVA tests with White’s correction for heteroscedastic data (comparing the relative abundances of males–females and water polo players–control group). Hypotheses were tested at the 5% level of random error.

## 3. Results

In general, *Streptococcus*, *Veilonella*, and *Prevotella* genera constituted more than 50% of the oral microbiome community of the studied youths aged between 16 and 18 years (Table 1).

Figure 1 and Figure 2 show the bacterium contents at the genus level by sexes and in water polo players and the control group, respectively. Visually, a difference could be detected between male and female subjects’ bacterial flora and between water polo players’ and control group subjects’ microbiome profiles. The frequent predominance of *Streptococcus* and *Veillonella* genera was visually greater in females vs. males and in water polo players vs. controls. To confirm this assumed difference between the subgroups, principal coordinate analysis of the abundances of the most frequent taxa was carried out on the genus level.

Principal coordinate analysis of the oral microbiome composition of subjects confirmed that the microbial profile had significant sexual dimorphism and differed between water polo players and children in the control group (Figure 3 and Figure 4, Table 2). The data points showed an apparent pattern of group-specific clustering in both by considering both the sex of the subjects (Figure 3) and the water polo players–control group comparison (Figure 4).

Compared to females, males had significantly higher relative abundance of the following genera (Table 3): *Atopobium* (1.6% vs. 1.0%, medium effect size), *Pravotella*_7 (20.4% vs. 12.2%, very large effect size, Table 3). Compared to females, males had significantly lower relative abundance of the genera: *Fusobacterium* (1.9% vs. 3.1%, large effect size), *Gemella* (3.4% vs. 5.0%, large effect size), and *Streptococcus* (19.3% vs. 25.3%, large effect size) (Wilcoxon signed rank test, *p* < 0.05). Compared to the control group, water polo players had significantly higher relative abundance of the genus *Veillonella* (18.3% vs. 14.6%, medium effect size) and significantly lower relative abundance of the genus *Gemella* (3.8% vs. 4.9%, large effect size).

## 4. Discussion

Past microbiological studies reported that the composition of the oral microbiome are characterized by great diversity and high proportions of anaerobic bacteria [29,30]. Overall, species of *Fusobacterium*, *Gemella*, *Granulicatella*, *Neisseria*, *Prevotella*, *Streptococcus*, and *Veillonella* were commonly detected in the oral microbiome of humans [30,31].

The development and changes in the oral microbiome throughout the human lifespan have not been fully explored. It is well known that the structure of the oral microbiome community is quite uniform during infancy and becomes gradually more complex by the appearance of new species (stages: predentate imprinting, eruption of primary teeth) in childhood. Puberty (and pregnancies) are also accompanied by significant changes in the oral microbial community due to changes in sex hormone levels. The composition and proportions of the oral microbes are quite stable in adulthood. It has been confirmed that the relative abundance of genera of *Enterobacterium*, *Pseudomonas*, *Staphylococcus*, and *Candida* increase with age [32].

The presented results revealed sexual dimorphism of the microbial composition of the oral microbiome in youth. Only the changes in the composition of the oral microbial community by age has been studied; sexual dimorphism in oral microbial communities has not been examined. The analyses revealed that *Prevotella_7*, *Streptococcus*, *Fusobacterium*, and *Atopobium* genera differed in their relative abundances (*p* < 0.05, with at least medium effect size, in the order of decreasing effect size) between the two sexes. Higher abundance of Prevotella (pathogen of supragingival plaque and subgingival plaque) and *Atopium* (a sexually transmitted pathogen of bacterial vaginosis and vaginitis—males cannot develop bacterial vaginosis but can spread the infection) and lower abundance of *Streptococcus* (pathogen of streptococcal disease) and Fusobacterium (pathogen of gingivitis, tonsillitis) genera characterized the males compared to females in the studied sample of youths.

Wang et al. [33] reported that *Streptococcus*, *Veillonella*, *Neisseria*, and *Haemophilus* were the most abundant genera (20%, 15%, 16%, 10%) in adults aged between 20 and 50 years (in saliva samples). Mashima et al. [34] found that *Streptococcus*, *Veillonella*, *Prevotella*, *Gemella*, and *Porphyromonas* genera (32%, 20%, 8%, 6%, 7%) comprised more than 70% of the microbial community in children aged between 7 and 15 years (with moderate oral hygiene). Takeshita et al. [35] found that the proportion of *Streptococcus*, *Veillonella*, and *Prevotella* genera was higher than 70% in elderly people aged 65 and over (Figure 5). The microbial analysis of studied 16–20-year-old youths revealed that more than 55% of the microbial community was constituted by *Streptococcus*, *Veillonella*, and *Prevotella*. These subjects’ oral microbiome was more similar to the microbial community of adults than children’s microbiome.

Water polo players had higher relative abundance of the genus *Veillonella* (pathogens of oral infections, *p* < 0.05, medium effect size), and lower relative abundance of the genus *Gemella* (pathogens of localized and generalized infections, especially in immunocompromised patients, *p* < 0.05, large effect size).

The special activity environment of water polo players (chlorinated water) prompted us to examine microbes that might be unique in water polo players. Corynebacterium argentoratense (*n*: 3 subjects, infections in the throat, respiratory tract) [36], Fusobacterium naviforme (*n*: 3 subjects, gingivitis) [37], *Howardella ureilytica* (*n*: 7 subjects, observed in humans in connection with decreased food consumption) [38], *Peptoniphilus lacrimalis* (*n*: 3 subjects, vaginal infections) [39], *Prevotella marshii* (*n*: 3 subjects, supragingival plaque, subgingival plaque) [40] were found only in the water polo players’ saliva samples. Most of these microbes can cause infections in the oral cavity, respiratory system, and vagina or inflammations in the gingiva and contribute to plaque formation. Only the *Howardella ureilytica* species was not included in this line of pathogens, as the presence of this microbe was found to be related with digestion since it is a plant substrate-degrading bacteria and has not been found to be pathogenic to humans.

Very few studies on the oral microbiome of athletes who train in swimming pool waters have been carried out until now. D’Ecole et al. found that cariogenic bacteria in swimmers were present in their oral microbiome with lower frequency than the “protective” microorganisms such as *S. sanguinis* [41]. *Streptococcus* species were not identified during the analysis, but the relative abundance of *Streptococcus* genera did not differ in the studied subsamples of water polo players and non-athletes.

Rowland et al. found a higher relative abundance of NO_3_-reducing bacterial genera in the oral microbiome (*Veillonella*, *Neisseria*, *Prevotella*, *Actinomyces*, *Rothia*, *Granulicatella*, *Staphylococcus*, *Propionibacterium*, and *Haemophilus*) after the pool-based training of swimmers than before the training [42]. The role of oral microflora as catalyzer in the reduction of NO_3_ ion could be one of the reasons for this observation. The increased level of *Veillonella* abundance in water polo players compared to the control group coincides with the results of Rowland and her colleagues.

Sport type-specific injuries and illnesses have already been explored and explained by the variability of environmental factors, extreme conditions, force loadings of the bones, joints, and the muscles, contact mechanisms, etc. These findings can help the evaluation of targeted sport-specific prevention strategies. Furthermore, the results of the presented analyses extended our knowledge in sports medicine from a microbial aspect: the most important implication of the results of the analysis was that regular screening for upper respiratory tract infections and oral hygiene (e.g., plaques, gingivitis) should be carried out among athletes who train in swimming pool waters and, among female athletes of water sports, for vaginal infections. This could improve the general health status of athletes in water sports, and, through this influence, it could have positive impact on their sport performance as well.

## 5. Conclusions

A summary of the microbial analyses in 16–20-year-old youths is as follows:The microbial profile of youths corresponded to the microbiome reported by previous studies during development of children and aging;Our data revealed a significant sexual dimorphism in the microbial community composition: females were found to develop streptococcal disease and tonsillitis more frequently than males, while males were found to have a higher abundance of pathogens of supragingival plaque, subgingival plaque, and a sexually transmitted disease, bacterial vaginosis;Water polo players’ oral microbiome differed from age-matched non-athletes;The dominant genera were present in both groups’ microbiome but in different proportions. Some genera were missing from the non-athlete age-peers’ microbiome, and those genera cause inflammations in the oral cavity, respiratory system, and vagina. This result suggests that regular water training can unfavorably alter the composition of the oral microbial community and can result in an increase in the abundance of inflammation-causing pathogens in particular, causing, unsurprisingly, oral infections.

## 6. Limitations of the Study

The main limitations of this study were financial limitations; therefore, only a specific number of saliva samples were examined. A longitudinal study could more precisely explore the relationship between oral microbial composition of youths and its influencing factors, such as lifestyle factors, health status, etc. The research data could not confirm that the oral microbiome of water polo players was exclusively characterized by the aquatic environment. It was not possible to collect samples from the swimming pools where the water polo players had their training. The only fact that could be acquired was that, according to the 37/1996. (X. 18.) NM Government Decree, “If a swimming pool hosts special sport events or competitions, additional water quality testing may be required before and after these events to ensure optimal water conditions for athletes”.

## Figures and Tables

**Figure 1 sports-11-00216-f001:**
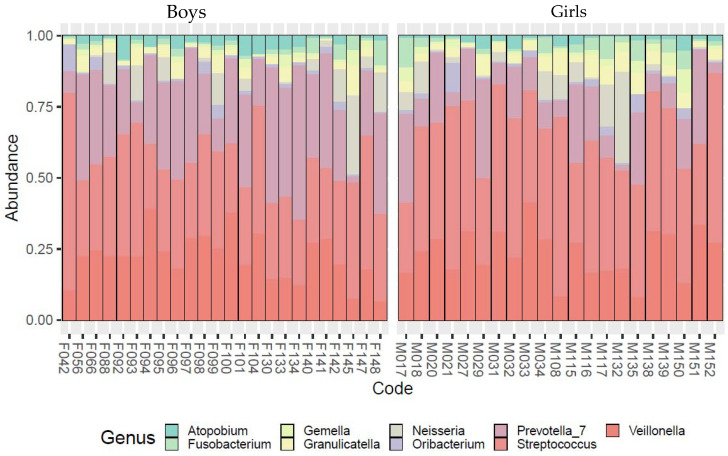
Bacterial community profiling of saliva samples by normalized abundances of the most frequent taxa on genus level (taxonomy rank abundances, only taxa with a relative abundance above 0.01 are shown) by sexes.

**Figure 2 sports-11-00216-f002:**
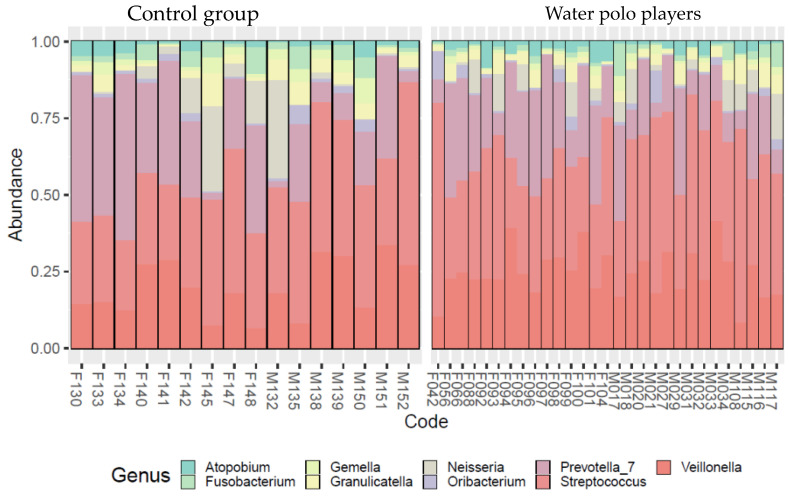
Bacterial community profiling of saliva samples by normalized abundances of the most frequent taxa on genus level (taxonomy rank abundances) in water polo players and control group.

**Figure 3 sports-11-00216-f003:**
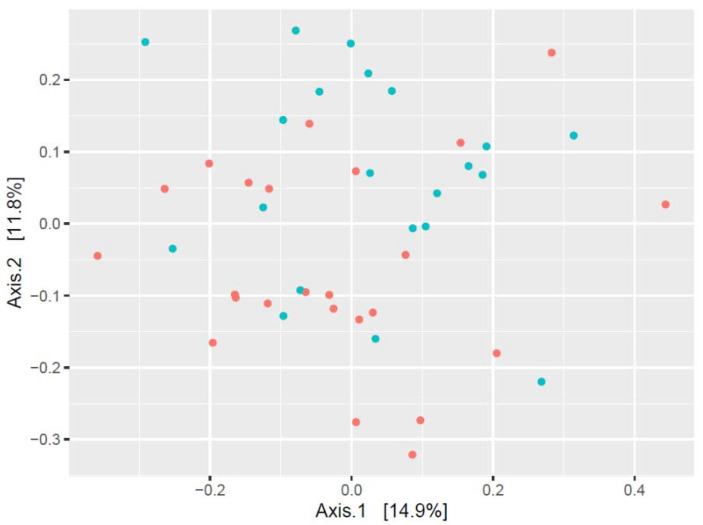
Beta diversity plot (method: principal coordinate analysis; distance calculation was carried out using Bray–Curtis method) by sexes (●: males, ●: females).

**Figure 4 sports-11-00216-f004:**
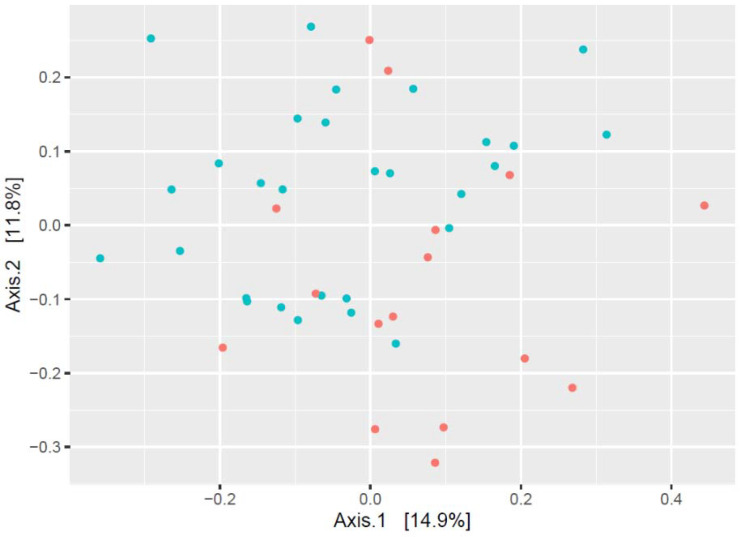
Beta diversity plot (method: principal coordinate analysis; distance calculation was carried out using Bray–Curtis method) in water polo players (●) and subjects belonging to the control group (●).

**Figure 5 sports-11-00216-f005:**
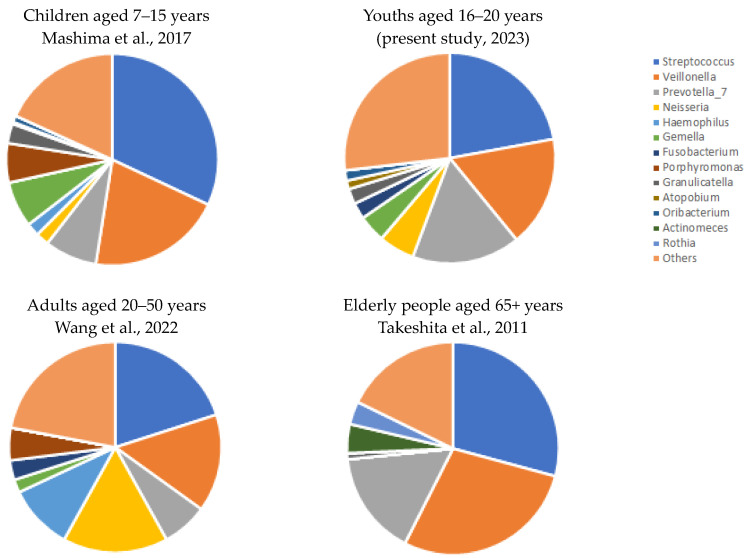
Mean genus abundances in the lifecycle stages in humans (sources: Takeshita et al., 2011 [28]; Mashima et al., 2017 [27]; Wang et al., 2022 [26]; and the present analysis).

**Table 1 sports-11-00216-t001:** Mean relative abundance of the most frequent bacteria in the oral microbiome in the studied sample of youth aged between 16 and 18 years.

Bacterial Genus	%
*Atopobium*	1.3
*Fusobacterium*	2.5
*Gemella*	4.2
*Granulicatella*	2.4
*Neisseria*	5.5
*Oribacterium*	1.6
*Prevotella_7*	16.6
*Streptococcus*	22.1
*Veillonella*	17.0

**Table 2 sports-11-00216-t002:** Summary of permutational multivariate analysis of variance (PERMANOVA) examining differences in microbiome profile based on the 16S rRNA gene for total bacterial and amplicon sequence variants’ bacterial degraders.

	Df	Sums Sqs	Mean Sqs	F Model	R^2^	*p*
Sex	1	0.308	0.308	1.883	0.040	0.016
Water polo players vs. control group	1	0.466	0.466	2.850	0.061	<0.001
Residuals	42	6.861	0.163	NA	0.899	NA
Total	44	7.634	NA	NA	1.000	NA

Df: degrees of freedom; Sum sqs: sum of squares, Mean sqs: mean squares, F: variance ratio, *p*: level of significance; values in bold indicate significant effects.

**Table 3 sports-11-00216-t003:** Significance level and effect size of analysis of the variance (ANOVA) by sex (M–F) and water polo players–control group comparison (W–C), respectively, on the relative abundance of the 16S rRNA gene OTUs identified at the genus level of bacteria.

Bacterium Genus	M–F	Effect Size	W–C	Effect Size
*Atopobium*	0.041	0.5 (medium)	0.248	0.2 (small)
*Fusobacterium*	0.029	0.6 (large)	0.919	0.2 (small)
*Gemella*	0.025	0.6 (large)	0.040	0.6 (large)
*Granulicatella*	0.262	0.3 (medium)	0.836	0.2 (small)
*Neisseria*	0.722	0.1 (small)	0.760	0.2 (small)
*Oribacterium*	0.709	0.1 (small)	0.494	0.2 (small)
*Prevotella_7*	0.001	1.0 (very large)	0.510	0.1 (small)
*Streptococcus*	0.002	0.8 (large)	0.426	0.1 (small)
*Veillonella*	0.257	0.3 (medium)	0.046	0.3 (medium)

## Data Availability

The datasets generated during and/or analyzed during the current study are available from the corresponding author on reasonable request.

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
