# Peer review of "The Oral Microbiome Profile of Water Polo Players Aged 16–20"

_sports, 2023, doi:10.3390/sports11110216_

Round 1
Reviewer 1 Report
Comments and Suggestions for Authors
This study aimed to analyze the composition of the oral microbiome of elite water polo players and compare it to the respective of non-athletes.
In my opinion, this study has major flaws throughout the text. More details are described below:
Abstract
· The 1st sentence does not match here. Please, use another Introductory sentence, more related to the content of the manuscript (e.g., athletes' oral health and its association with sports performance).
· The research hypothesis is not mentioned in the Abstract. It is usually placed at the end of the Introduction part.
· Please, provide some statistical values in the Abstract (e.g., p values).
· Why did the authors come up to this conclusion? It is very generic and not strictly related to the purpose and the findings of the study.
Introduction
· L. 59-63. Please enlarge this sentence.
· Why is it important to compare the oral health of water polo athletes with no athletic populations? There is no rationale for conducting this study.
· The research hypothesis is missing here.
· Recent and related references are absent in this part (e.g., Tripodi, et al. The Impact of Sport Training on Oral Health in Athletes. Dent. J. 2021, 9, 51. https://doi.org/ 10.3390/dj9050051).
Statistical analysis
· This part is also problematic – for instance, no specific tests for post-hoc analysis are reported.
Discussion
· This part is completely missing! No justification for the Results, no discussion with previous findings, and no Limitations part!
Comments on the Quality of English LanguageMinor English check is required.
Author Response
General Comment: This study aimed to analyze the composition of the oral microbiome of elite water polo players and compare it to the respective of non-athletes.
In my opinion, this study has major flaws throughout the text. More details are described below:
Response: The Authors would like to express sincere thanks to the Reviewer for careful reading and suggestion for improvement in the paper. The replies to the suggestions and comments are presented in the order of the Reviewer’s comments.
Abstract
Comment 1: The 1st sentence does not match here. Please, use another Introductory sentence, more related to the content of the manuscript (e.g., athletes' oral health and its association with sports performance).
Response: The Authors thank this suggestion of the Reviewer.
- Accepting the reviewer's suggestion, the 1st sentence was changed as follow: Clorine has a strong antibacterial property and is the most common disinfectant that is used in swimming pools. Therefore, the microbiota community in the oral cavity of those who practice water sports is assumed to be special due to their regular participation in water.
Comment 2: The research hypothesis is not mentioned in the Abstract. It is usually placed at the end of the Introduction part.
Response: The Authors thank this comment of the Reviewer.
-The research hypothesis was placed at the end of the Introduction part as follows: We hypothesized that the oral cavity microbiota community differed between water polo players and non-athletes.
Comment 3: Please, provide some statistical values in the Abstract (e.g., p values).
Response: The Authors thank this comment as well, the Abstract was completed with p values in the Results section.
Comment 4: Why did the authors come up to this conclusion? It is very generic and not strictly related to the purpose and the findings of the study.
Response: The Authors thank this remark of the Reviewer.
- Based on reviewer's comment, the conclusion was changed as follow: The results suggest that regular water training can unfavorably alter the composition of the oral microbial community.
Introduction
Comment 5: L. 59-63. Please enlarge this sentence.
Response: The Authors thank this suggestion of the Reviewer.
- Accepting the reviewer's suggestion, the sentence in L. 59-63. was enlarged as follow: L.67-73: In addition to the significant influence of host genetics on the composition of the salivary microbiome, it has been distinctly observed that lifestyle factors play an even more substantial role in shaping the diversity of the salivary microbiome [4]. Factors such as age, gender, body composition, ethnic background, geographical location, sociodemographic variables, as well as habits such as smoking, and alcohol consumption have all been demonstrated to exert a discernible impact on the composition of the oral microbiome [5-12].
Comment 6: Why is it important to compare the oral health of water polo athletes with no athletic populations? There is no rationale for conducting this study.
Response: The Authors thank this question of the Reviewer. Our aim was to compare the composition of the oral microbiome in youth water polo players with healthy Hungarian peers. We chose non-athletes as our control group because sports activities can also impact the composition of the oral microbiome community.
Comment 7: The research hypothesis is missing here.
Response: The Authors thank this comment of the Reviewer.
- Accepting the reviewer's comment, the research hypothesis was placed in the Introduction part as follows: We hypothesized that the oral cavity microbiota community differed between water polo players and non-athletes due to training in chlorinated water.
Comment 8: Recent and related references are absent in this part (e.g., Tripodi, et al. The Impact of Sport Training on Oral Health in Athletes. Dent. J. 2021, 9, 51. https://doi.org/ 10.3390/dj9050051).
Response: The Authors thank this comment of the Reviewer as well.
The reference recommended by the reviewer was included in the introductory part as follows: - Line 80-83. The research carried out by Tripodi et al. has also confirmed that among athletes participating in various sports, engaging in sports activities can be regarded as a potential risk factor for the development of oral diseases.
Statistical analysis:
Comment 9: This part is also problematic – for instance, no specific tests for post-hoc analysis are reported.
Response: The Authors thank this comment as well. The compositional similarities between the different groups were studied with permutational multivariate analysis of variance analysis and for the differential abundance testing the univariate analysis of relative abundance of genera was performed with ANOVA test with White’s correction for heteroscedastic data (comparing the relative abundances of males-females, water polo players-control group). Two subgroups (males-females, water polo players-control group) were compared in the all the analyses, not more than 2 subgroups were selected for the analyses, that is why post-hoc analyses were not used. The mean relative abundance of the most frequent bacteria in the oral microbiome was presented but not tested for differences among the bacterial genera, neither ANOVA, nor post-hoc tests were performed in this part of the analysis.
Discussion
Comment 10: This part is completely missing! No justification for the Results, no discussion with previous findings, and no Limitations part!
Response: The Authors are grateful for this comment as well, the Conclusions section contained statements that fit better the Discussion section, therefor Conclusions section was modified to Discussion section and a new Conclusions section was inserted in the manuscript as follows:
“As a summary of the microbial analyses in 16-20-year-old youths is as follows:
- the microbial profile of youths corresponded to the microbiome reported also by previous studies during development of children and aging;
- our data revealed a significant sexual dimorphism in the microbial community composition – females were found to develop streptococcal disease, tonsillitis more frequently than males, while males were found to higher abundance of pathogens of supragingival plaque, subgingival plaque and a sexually transmitted disease, bacterial vaginosis;
- water polo players’ oral microbiome differed from age-matched non-athletes:
- the dominant genera were present in both groups’ microbiome but in different proportions, but had some genera that were missing from the non-athlete age-peers’ microbiome, and that genera cause inflammations in the oral cavity, respiratory system, vagina. This result suggests that regular water training can unfavorably alter the composition of the oral microbial community, can result in the increase of abundance of inflammation-causing pathogens, especially and unsurprisingly causing oral infections.”
Reviewer 2 Report
Comments and Suggestions for Authors
In my opinion this manuscript, despite providing some interesting data about the oral microbiota composition in waterpolo players, might require some improvements I hope I can support with these suggestions:
- Abstract: specify if only males were included; provide statistics (p values and effect size) for the proposed differences between the groups; provide some practical indication regarding the findings in the conclusion.
- Introduction:
Lines 98-102: is this really necessary? It does not seem to provide any additional information to study. I suggest to remove it or to provide a better link to the hypothesis and rationale for this study.
- Methods:
Subjects, specify please how participants were recruited, and the inclusion and exclusion criteria; referring to "those who participated in the body structural examinations" is not sufficient, and it does not refer to any other publication.
Table 1 might not be necessary, it is not well understandable and does not provide any helpful information for this study
- Discussion:
Do the authors think that hydration habits could affect the results both in terms of general habits (during typical training) and before data collection? For example, it has been suggested that poor fluid intake might affect gut microbiota (Vanhaecke et al., J Nutr, 2022) and oral health (Kim, Water, 2021), and waterpolo as well as swimmers might be characterized by peculiar water intake during training due to the specific sport characteristics (Cox et al., J Sci Med Sport, 2002; Buoite Stella et al., J Sports Med Phys Fitness, 2017). Maybe it might be worth discussion.
Author Response
The Authors would like to express sincere thanks to the Reviewer for careful reading and suggestion for improvement in the paper.
General Comment: In my opinion this manuscript, despite providing some interesting data about the oral microbiota composition in water polo players, might require some improvements. I hope I can support with these suggestions:
Response: We are grateful for the general comment of the Reviewer, the replies to the suggestions and comments are presented in the order of the Reviewer’s comments.
Abstract
Comment 1: Specify if only males were included; provide statistics (p values and effect size) for the proposed differences between the groups; provide some practical indication regarding the findings in the conclusion.
Response: The Authors thank this comment of the Reviewer. The manuscript was completed with p values and effect sizes of the tests used to compare subgroups (males – females, water polo players – control group) in Table 3 and in Results and Discussion sections, as well as in the Abstract.
Introduction
Comment 2: Lines 98-102: is this really necessary? It does not seem to provide any additional information to study. I suggest to remove it or to provide a better link to the hypothesis and rationale for this study.
Response: The Authors thank this suggestion of the Reviewer as well. By accepting the reviewer's suggestion, the sentence in the lines 98-102 was removed.
Methods
Comment 3: Subjects, specify please how participants were recruited, and the inclusion and exclusion criteria; referring to "those who participated in the body structural examinations" is not sufficient, and it does not refer to any other publication.
Response: The Authors thank this suggestion of the Reviewer. The inclusion and exclusion criteria for the collection of saliva samples met the ethical standards of the project. These criteria we were pre-defined in the study protocol to ensure that the collected saliva samples were relevant to our research objectives. The Study design section was completed with the criteria as follows:
Inclusion criteria of the participants were: Age; Gender; Training regime (in the case of the control group, an inactivity time); Health status; Medication use; Time since last food or drink; Smoking abuse; Specific medical conditions; Pregnancy status.
Exclusion criteria of the participants were: Communicable diseases (HIV, hepatitis or COVID-19); Recent dental work; Medications that alter saliva (anticholinergic medications); Serious medical conditions; Drug abuse; Allergies or sensitivities; Inability to provide consent; Participation in other studies; Ethical considerations.
Comment 4: Table 1 might not be necessary, it is not well understandable and does not provide any helpful information for this study.
Response: The Authors thank this comment, Table 1 was deleted from the manuscript.
Discussion
Comment 5: Do the authors think that hydration habits could affect the results both in terms of general habits (during typical training) and before data collection? For example, it has been suggested that poor fluid intake might affect gut microbiota (Vanhaecke et al., J Nutr, 2022) and oral health (Kim, Water, 2021), and waterpolo as well as swimmers might be characterized by peculiar water intake during training due to the specific sport characteristics (Cox et al., J Sci Med Sport, 2002; Buoite Stella et al., J Sports Med Phys Fitness, 2017). Maybe it might be worth discussion.
Response: The Authors thank this suggestion of the Reviewer. The manuscript was completed in the Introduction section with the followings:
“Hydration habits can potentially affect the composition of the oral microbiome. The oral microbiome is a complex community of microorganisms, including bacteria, viruses, fungi, and other microorganisms, that inhabit the oral cavity. These microorganisms play a crucial role in maintaining oral health and can influence overall health as well. Hydration status directly affects saliva production. Saliva contains antimicrobial proteins and enzymes that help control the growth of harmful bacteria in the mouth. When you are well-hydrated, your salivary flow is typically normal, which can contribute to a healthier oral microbiome (Yu-Rin Kim, 2010). Saliva helps maintain the pH balance in the mouth. Adequate hydration can support the buffering capacity of saliva, preventing the oral environment from becoming too acidic or too alkaline, which can impact the types of microorganisms that thrive in the mouth (Popkin at al., 2017). Dehydration can lead to dry mouth (xerostomia), which can be a breeding ground for harmful bacteria. When there's insufficient saliva to wash away food particles and neutralize acids, it can lead to an imbalance in the oral microbiome, allowing harmful bacteria to flourish (Perrier at al., 2013; Vanhaecke et al., 2022). Waterpolo as well as swimmers might be characterized by peculiar water intake during training due to the specific sport characteristics. There was a wide individual variation in fluid intake and sweat loss of both water polo players and swimmers. (Cox et al., 2002). Dehydration experienced by athletes in that study was less than typically reported for "land-based" athletes.
Unlike other team sports, few researchers have examined the water intake habits of water polo athletes which affect the athletes’ hydration. Overall, while hydration habits alone may not determine the composition of the oral microbiome, they can play a significant role in maintaining a healthy oral environment. Staying well-hydrated, practicing good oral hygiene, and making healthy dietary choices can all contribute to a balanced and beneficial oral microbiome.”
The References were completed with the cited publications:
Yu-Rin Kim. Analysis of the Effect of Daily Water Intake on Oral Health: Result from Seven Waves of a Population-Based Panel Study. Water 2021, 13(19), 2716. doi: https://doi.org/10.3390/w13192716
Vanhaecke, T.; Bretin, O.; Poirel, M.; Tap, J. Drinking Water Source and Intake Are Associated with Distinct Gut Microbiota Signatures in US and UK Populations. J Nutr. 2022, 152(1):171-182. doi: https://doi.org/10.1093/jn/nxab312
Perrier, E.; Vergne, S.; Klein, A.; Poupin, M.; Rondeau, P.; Le Bellego, L.; Armstrong, L.E.; Lang, F.; Stookey, J.; Tack, I. Hydration biomarkers in free-living adults with different levels of habitual fluid consumption. Br. J. Nutr. 2013, 109, 1678–1687. doi: https://doi.org/10.1017/S0007114512003601
Popkin, B.M.; D’Anci, K.E.; Rosenberg, I.H. Water, hydration, and health. Nutr. Rev. 2010, 68, 439–458. doi: https://doi.org/10.1111/j.1753-4887.2010.00304.x
Cox, G.R.; Broad, E.M.; Riley, M.D.; Burke, L.M. Body mass changes and voluntary fluid intakes of elite level water polo players and swimmers. J Sci Med Sport. 2002, 5(3):183-93. doi: https://doi.org/10.1016/s1440-2440(02)80003-2.

Reviewer 3 Report
Comments and Suggestions for Authors
My recommendations are the following:
There are major differences between the number of subjects presented in the abstract compared to the Subjects section. I recommend the correction. Age is also uncorrelated.
I recommend that in the Subjects section you mention the inclusion and exclusion criteria.
Lines 116-117 at the age of 20 are no longer considered children, I recommend the correction. Possibly use the subject word. I recommend that you mention the standard deviation in parentheses for the age.
Figure 3 mentions what the axes represent.
I recommend rewriting and reorganizing the Conclusions section into two subsections, namely: Discussions and Conclusions
I recommend that the Discussion section be expanded with new correlations between the results of this study and results from previous studies.
I recommend mentioning the limitations of this study.
I consider that this confirmatory study does not have a major relevance. If a comparative analysis of the microflora was carried out depending on the duration of exposure in the water, it was more specific and relevant by age category. It is not clear from the study if this microflora is due exclusively to the aquatic environment.
Author Response
The Authors would like to express sincere thanks to the Reviewer for careful reading and suggestion for improvement in the paper. The replies to the suggestions and comments are presented in the order of the Reviewer’s comments.
Comment 1: There are major differences between the number of subjects presented in the abstract compared to the Subjects section. I recommend the correction. Age is also uncorrelated.
Response: The Authors thank this suggestion of the Reviewer.
- Accepting the reviewer's suggestion, the number of subjects presented in the abstract was correction as follow:
“Altogether 124 water polo players (62 males and 62 females, aged between 9-20 years) and 16 non-athlete youths (control group, 8 males and 8 females, aged between 16-20 years, mean age +SD = 17.1 + 1.4 years) who participated in the body structural examinations voluntarily agreed to participate in the study. In a randomly selected subsample of water polo players (n: 29, aged between 16-20 years, mean age +SD = 17.3 + 1.0 years) saliva samples were also collected. Saliva samples were collected from all non-athlete youths (n: 16, aged between 16-20 years).”
Comment 2: I recommend that in the Subjects section you mention the inclusion and exclusion criteria.
Response: The Authors thank this suggestion of the Reviewer. The inclusion and exclusion criteria for the collection of saliva samples met the ethical standards of the project. The Study design section was completed with the criteria as follows:
“Inclusion criteria of the participants were: Age; Gender; Training regime (in the case of the control group, an inactivity time); Health status; Medication use; Time since last food or drink; Smoking abuse; Specific medical conditions; Pregnancy status.
Exclusion criteria of the participants were: Communicable diseases (HIV, hepatitis or COVID-19); Recent dental work; Medications that alter saliva (anticholinergic medications); Serious medical conditions; Drug abuse; Allergies or sensitivities; Inability to provide consent; Participation in other studies; Ethical considerations.”
Comment 3: Lines 116-117 at the age of 20 are no longer considered children, I recommend the correction. Possibly use the subject word. I recommend that you mention the standard deviation in parentheses for the age.
Response: The Authors thank this suggestion of the Reviewer.
- The mean and standard deviation of age for the two subgroups (athletes and non-athletes) was inserted in the sample description (Response to Comment 1 contains this supplementation).
- Accepting the reviewer's suggestion, the lines 116-117 was changed as follows:
L.143-144:“Saliva samples were collected from all non-athlete youths (n: 16, aged between 16-20 years).”
Comment 4: Figure 3 mentions what the axes represent.
Response: The Authors thank this comment as well. Axes 1 and 2 are Principal Coordinates 1 and 2. The percentages are the percent of total variation explained by that axis. Principal Coordinate Analysis Plot (PCoA) is used for visualization of the data in the distance matrix in a plot. The distance matrix is transformed into a new set of orthogonal axes where the first axis (PC1) can be used to explain the maximum amount of variation present in the dataset, followed by the second axis (PC2), third (PC3), etc. The axis labels are values for % variability explained by each principal coordinate (PCo) axis. In the case of the presented analysis, PC1 and PC2 values were sufficient to visualize the 2-2 subgroups (males-females, water polo players - non-athletes) by considering the microbiome relative abundances. The captions of Figures 3-4 were completed with the following explanations:
“In the Principal Coordinate Analysis the distance matrix was transformed into a new set of orthogonal axes – Axis.1 (PC1) and Axis.2, axes can explain the maximum amount of variation present in the dataset by descending order, the axis labels represent % of variability explained by each PCo axis in the Figures.”
Comment 5: I recommend rewriting and reorganizing the Conclusions section into two subsections, namely: Discussions and Conclusions.
Response: The Authors thank this suggestion of the Reviewer. By accepting the Reviewer’s suggestion, the Conclusions section was rewritten and reorganized into two subsections: Discussion and Conclusions.
Comment 6: I recommend that the Discussion section be expanded with new correlations between the results of this study and results from previous studies.
Response: The Authors thank this comment of the Reviewer as well. The Discussion section was completed as follows:
“Very few studies on the oral microbiome of athletes having training in swimming pool waters have been carried out until now. D’Ecole et al. found in swimmers that cariogenic bacteria were present in their oral microbiome with lower frequency than the “protective” microorganisms such as S. sanguinis (2016). Streptococcus species were not identified during the analysis, but the relative abundance of Streptococcus genera did not differ in the studied subsamples of water polo players and non-athletes.
Rowland et al. found higher relative abundance of NO3-reducing bacterial genera in the oral microbiome (Veillonella, Neisseria, Prevotella, Actinomyces, Rothia, Granulicatella, Staphylococcus, Propionibacterium and Haemophilus) after the pool-based training of swimmers than before the training. The role of oral microflora as catalyzer in the reduction of NO3 ion could be one of the reasons for this observation. The increased level of Veillonella abundance in water polo players than in the control group coincides with the results of Rowland and her colleagues.”
The References were supplemented with the cited publications:
D’Ercole, S.; Tieri, M.; Martinelli, D.; Tripodi, D. The effect of swimming on oral health status: Competitive versus non-competitive athletes. J. Appl. Oral Sci. 2016, 24, 107–113.
Rowland, S.N., Chessor, R., French, G., Robinson, G.P., O’Donnell, E., James, L.J. and Bailey, S.J., 2021. Oral nitrate reduction is not impaired after training in chlorinated swimming pool water in elite swimmers. Applied Physiology, Nutrition, and Metabolism, 46(1), pp.86-89.
Comment 7: I recommend mentioning the limitations of this study.
Response: The Authors thank this suggestion of the Reviewer. The Limitations section was inserted in the manuscript as follows:
“The main limitations of this study were financial limitations, therefore only a specific number of saliva samples were examined. A longitudinal study could more precisely explore the relationship between oral microbial composition of youths and its influencing factors as lifestyle factors, health status, microbial composition of the swimming pool occupied by the athletes, etc.”
Comment 8: I consider that this confirmatory study does not have a major relevance. If a comparative analysis of the microflora was carried out depending on the duration of exposure in the water, it was more specific and relevant by age category. It is not clear from the study if this microflora is due exclusively to the aquatic environment.
Response: Response: The Authors thank this suggestion of the Reviewer.
The microflora of a swimming pool primarily consists of microorganisms that are introduced through various sources, including the water supply, swimmers, and environmental factors. Swimming pools are typically treated with chemicals, such as chlorine, to maintain water quality and inhibit the growth of harmful microorganisms. As a result, the microbial community in a properly maintained swimming pool is generally limited, and the water is expected to be safe for recreational use. While chlorine is effective at disinfecting swimming pool water, some bacteria, particularly certain species of Mycobacterium, can be resistant to chlorine. These bacteria can potentially survive and persist in pool water, although they are usually present in low numbers. Certain protozoa, such as Cryptosporidium and Giardia, are chlorine-resistant too and can cause gastrointestinal illnesses if ingested through contaminated pool water. Microorganisms can form biofilms on pool surfaces, including pool walls, floors, and filtration systems. Biofilms can provide a protective environment for bacteria, making them more resistant to disinfection.
It's important to note that swimming pool water quality is closely monitored and regulated to ensure the safety of swimmers. Public swimming pools are subject to health and safety standards that dictate water quality parameters, including chlorine levels, pH, and microbial contamination. Regular water testing and maintenance are essential to prevent waterborne illnesses and to keep the pool water clean and safe for recreational use. A swimming pools, in particular, are subject to regulations and health codes that specify water quality standards and safety requirements. A sport swimming pools in Hungary, which are subject to stricter regulations, often have a more frequent testing schedule. Water quality in pools is usually tested multiple times a day, with parameters like chlorine levels, pH, alkalinity, and temperature being monitored regularly. The exact frequency may vary based on pool size, usage, and local regulations. If a swimming pool hosts special sport events or competitions, additional water quality testing may be required before and after these events to ensure optimal water conditions for athletes (37/1996. (X. 18.) NM government decree).
The Limitations section was completed with the followings:
“… The research data could not confirm that the oral microbiome of water polo players was exclusively characterized by the aquatic environment. It was not possible to collect samples from the swimming pools where the water polo players had their trainings. The only fact that could be acquired was that according to the 37/1996. (X. 18.) NM Government Decree “If a swimming pool hosts special sport events or competitions, additional water quality testing may be required before and after these events to ensure optimal water conditions for athletes.”

Round 2
Reviewer 1 Report
Comments and Suggestions for Authors
The authors did a quite good job of revising the manuscript in line with my concerns. Nevertheless,
the manuscript requires further major revisions before becoming acceptable for publication, mainly
in the Discussion part, which still remains incomplete, as the authors still fail to discuss the possible
implications for coaches/researchers/nutritionists/and - or sports organizations

Author Response
The Authors would like to express sincere thanks again to the Reviewer for careful reading and the comment of the 2nd review for improvement in the paper.
Comment: The authors did a quite good job of revising the manuscript in line with my concerns. Nevertheless, the manuscript requires further major revisions before becoming acceptable for publication, mainly in the Discussion part, which still remains incomplete, as the authors still fail to discuss the possible implications for coaches/researchers/nutritionists/and - or sports organizations.
Response: The Authors thank this comment of the Reviewer. The manuscript was corrected by following the Reviewer’s suggestion in the Discussion section as follows:
“Sport type-specific injuries and illnesses have been already explored and explained by the variability of the environmental factors, extreme conditions, force loadings of the bones, joints, and the muscles, contact mechanisms, etc. These findings can help the evaluation of targeted sport-specific prevention strategies. Furthermore, the results of the presented analyses extended our knowledge in sports medicine from a microbial aspect: the most important implication of the results of the analysis was that regular screening for upper respiratory tract infections, oral hygiene (e.g. plaques, gingivitis) should be carried out among athletes having trainings in swimming pool waters, and for vaginal infections in female athletes of water sports. This could improve the general health status of athletes in water sports, and through this influence it could have positive impact on their sport performance as well.”

Reviewer 3 Report
Comments and Suggestions for Authors
No comments
Author Response
The Authors would like to express sincere thanks to the Reviewer for the careful reading of the revised manuscript. The 2nd review of Reviewer (3) does not contain any comment or suggestion to be replied in the submission surface in the 'author view', the manuscript revised by following the comment of the other Reviewer (1) in the Discussion section.